# Age-Related Changes in Postural Stability in Response to Varying Surface Instability in Young and Middle-Aged Adults

**DOI:** 10.3390/s24216846

**Published:** 2024-10-25

**Authors:** Arunee Promsri, Punnakan Pitiwattanakulchai, Siwaporn Saodan, Salinrat Thiwan

**Affiliations:** 1Department of Physical Therapy, School of Allied Health Sciences, University of Phayao, Phayao 5600, Thailand; 64130183@up.ac.th (P.P.); 64131195@up.ac.th (S.S.); 64130251@up.ac.th (S.T.); 2Department of Sport Science, University of Innsbruck, A-6020 Innsbruck, Austria

**Keywords:** postural control, body composition, perception of effort, middle age, fall risk prevention, sample entropy, surface instability, IMU sensor

## Abstract

As individuals transition into middle age, subtle declines in postural control may occur due to gradual reductions in neuromuscular control. The current study aimed to examine the effect of age on bipedal postural control across three support surfaces with varying degrees of instability: a firm surface, a foam pad, and a multiaxial balance board. The effect of surface stability was also assessed. Postural accelerations were recorded using a tri-axial accelerometer placed over the lumbar spine (L5) in 24 young female adults (23.9 ± 5.3 years) and 24 middle-aged female adults (51.4 ± 5.9 years). Sample entropy (SampEn) was used to analyze the complexity of postural control by measuring the regularity of postural acceleration. The main results show significant age-related differences in the mediolateral and anteroposterior acceleration directions (*p* ≤ 0.012). Young adults exhibit more irregular fluctuations in postural acceleration (high SampEn), reflecting greater efficiency or automaticity in postural control compared to middle-aged adults. Increased surface instability also progressively decreases SampEn in the mediolateral direction (*p* < 0.001), reflecting less automaticity with increased instability. However, no interaction effects are observed. These findings imply that incorporating balance training on unstable surfaces might help middle-aged adults maintain postural control and prevent future falls.

## 1. Introduction

Postural control is typically defined as the ability to maintain body posture in space, which is crucial for ensuring stability and orientation [1]. Maintaining posture is essential for daily activities and overall quality of life, as it involves keeping the body’s center of mass within its base of support to prevent falls and injuries, particularly among the elderly [2]. Effective postural stability depends on the integration of sensory feedback from vision, proprioception, and the vestibular system, all of which support balance and alignment [3].

As individuals age, declines in sensory perception, motor abilities, and cognitive function are consistently observed, contributing to reduced postural stability [4,5]. This decline manifests as decreased balance, slower reaction times, and an increased risk of falls [6,7]. Postural impairment is linked to age-related sensory deficits, with greater postural sway observed from young adulthood through middle and older adulthood [8]. Middle age, typically defined as ages 40 to 60 years [9], is a period where individuals generally demonstrate more stability than older adults, but these declines can still impact postural stability and coordination [4]. The prevalence of health-related issues, such as chronic illnesses, medication use, dizziness, and vision and hearing problems, may further exacerbate these declines [10]. For instance, middle age can worsen the effects of challenging acoustic conditions on postural control compared to young adults, as increased effort to process auditory information in adverse environments depletes cognitive resources needed for balance [11]. Understanding these changes is crucial for developing interventions to reduce fall risk and enhance quality of life during the transition to older adulthood. Moreover, the natural decline in sensory processing and neuromuscular coordination makes maintaining postural stability increasingly difficult [12]. These declines, further influenced by gender-related factors such as muscle mass distribution and hormonal differences, can make women particularly vulnerable to balance impairments [13]. When these age- and gender-related challenges are combined with the increased demands of maintaining balance on unstable surfaces, the risk of instability and falls is heightened, especially in middle-aged adults [14]. Gender differences also complicate postural control, with women at higher risk of falling due to factors like muscle and bone structure differences, hormonal influences, and social factors affecting activity levels and risk perception [15]. Addressing these gender-specific issues is essential for creating effective strategies to improve postural stability and prevent falls, especially among women [13].

When considering environmental factors affecting postural stability, surface stability significantly alters the demands placed on the postural control system [16]. Balancing on unstable surfaces requires greater sensory integration and neuromuscular coordination, potentially revealing age-related differences in balance capabilities [17]. Stable surfaces, such as hard floors, provide consistent sensory feedback, support balance with minimal neuromuscular effort, and allow optimal stability mechanisms [18]. In contrast, unstable surfaces, like foam pads or wobble boards, disrupt typical sensory feedback with uneven and unpredictable signals [19]. Previous research suggests that unstable surfaces challenge postural stability by reducing sensory input and diminishing the effectiveness of corrective ankle torque [20,21]. Understanding how surface stability impacts postural control and perceived effort is crucial for developing effective rehabilitation and exercise programs to improve stability and coordination across different populations.

Body accelerations can arise from muscle actions, the use of gravity to achieve desired acceleration, or unintended gravitational effects that the neuromuscular system must counteract [22]. Measuring body acceleration can, thus, effectively analyze human movements such as gait [23] and postural control [24,25]. Since postural control is thought to result from nonlinear interactions among various neuromuscular components and internal and external factors [26,27], nonlinear analyses, such as sample entropy (SampEn) [26,28,29,30,31,32], provide valuable insights into how these factors influence postural control by examining the temporal evolution of postural adjustments [33]. SampEn assesses the complexity of the postural control system by measuring the regularity or predictability of postural stability [34]. High SampEn values, which reflect greater irregularity, suggest a more adaptable and efficient postural control system, often associated with more automatic (less consciously effortful) balance control [29,32,35]. In contrast, low SampEn values indicate a more rigid or deliberate control strategy, reflecting less automatic and potentially less effective control [29,32,35]. However, excessively high SampEn values can indicate erratic movements and diminished postural control [34]. SampEn should ideally fall within a moderate range, as deviations may reflect reduced control effectiveness [34]. The current study uses SampEn to measure postural control complexity by assessing the regularity of postural sway acceleration, with lower values indicating less automatic and more consciously controlled balance [36]. This measure may help elucidate the effects of age-related changes in postural control and surface instability, serving as an indirect indicator of the perceived effort required to maintain balance.

In summary, this study investigates how age and surface stability affect postural stability by comparing postural control complexity, assessed through SampEn of body acceleration, between young and middle-aged individuals during bipedal balance tasks on surfaces with varying instability levels. We hypothesize that middle-aged individuals will exhibit reduced postural stability compared to younger individuals, particularly on unstable surfaces. This hypothesis is based on the understanding that age-related declines in neuromuscular function and balance control become more pronounced with middle age, making it more challenging to maintain stability on unstable surfaces [19]. Additionally, surface instability is known to disrupt postural control and increase neuromuscular demands [19], potentially exacerbating differences between age groups. Understanding these changes is crucial for developing targeted strategies to improve postural stability and prevent falls in middle-aged adults. We also test the correlations between body composition (e.g., percent body fat and percent muscle mass) and postural control complexity [37].

## 2. Materials and Methods

### 2.1. Participants

Forty-eight participants, consisting of 24 young adults and 24 middle-aged adults, took part in the study. All participants reported no neurological or musculoskeletal issues and had not undergone balance-specific training in the past six months. The study procedures were approved by the Institutional Review Board of the University of Phayao, Thailand (Approval Code No.: UP-HEC 1.2/031/67) and adhered to the Declaration of Helsinki. Written informed consent was obtained from all volunteers prior to their participation. The minimum sample size required for the study was determined using a priori power analysis with G*Power software version 3.1.9.4 (Heinrich-Heine-Universität Düsseldorf, Düsseldorf, Germany) [38], with parameters set at an effect size (partial eta squared) of η_p_^2^ = 0.25, α = 0.05, and 1-β = 0.95, which suggested a total sample size of *n* = 36. Nevertheless, 48 adults volunteered to take part in the study. Characteristics of participants are shown in Table 1.

### 2.2. Equipment and Experimental Procedure

Before carrying out balance tasks, the participants’ body height was measured using a portable Seca 217 (SECA GmbH, Hamburg, Germany), and the body composition (including total body mass, fat mass, lean body mass, and muscle mass) was measured using a bioimpedance analysis body composition analyzer (InBody 270, Biospace, Seoul, Republic of Korea). A previous study demonstrated strong correlations (*r* = 0.97–0.99) and relative (ICC = 0.98–1.00) inter-device reliability between the InBody 270 and dual-energy X-ray absorptiometry (DXA), indicating that the InBody 270 is a reliable tool for body composition analysis [39].

A tri-axial accelerometer (BWT901CL Bluetooth 2.0 9-axis IMU sensor, WitMotion Ltd., Dong Guan, China) was used to measure body acceleration. It was attached horizontally to the lumbar region (L5) with a waist-mounted belt, positioning it near the body’s center of mass (COM) [24,40]. Before the test, participants stood facing a wall. During the test, they were instructed to alternately flex their hips (i.e., perform marching) up to a marked line on the wall as quickly as possible for two minutes, with their hands relaxed at their sides. The accelerometer was operated via the “WitMotion” app, version 5.0.9, on iPhone 11 (Apple Inc., Los Altos, CA, USA), with a sampling rate of 200 Hz.

The study utilized three different support surfaces (see Figure 1): a firm surface, a foam pad (Airex Balance Pad Elite, Alcan Airex AG, Sins, Switzerland), and an MFT Challenge Disc (Trend Sport Trading GmbH, Großhöflein, Austria). The foam pad is a rectangular piece of blue foam, measuring 50 cm in width, 41 cm in length, and 6 cm in height. The MFT Challenge Disc is a multiaxial balance board with a 44 cm diameter circular plate on top, connected to a base plate by four 8 cm high rubber cylinders positioned at the center of the plate. To ensure a consistent starting position (Figure 1), participants were instructed to place their hands on their iliac crests, stand barefoot with the base of their second metatarsal bone aligned with a reticle crossline marked on each surface, and position the medial borders of their feet (the distal ends of the first metatarsal bones) in line with a specific inter-feet distance (15% of the biacromial diameter) [19].

For the measurements, each participant initially completed a 15 s familiarization trial on the foam pad and balance board without any specific instructions or feedback. Following this, participants performed three 120 s balance trials barefoot on each of the three surfaces: a firm surface, a foam pad, and an MFT Challenge Disc. The order of testing the surfaces was randomized. During the balance tasks, participants were instructed to focus their gaze on a 10 cm diameter black circle on the wall, positioned about two meters away at their eye level, and avoid unnecessary movements, such as scratching [41]. For the firm and foam surfaces, participants were required to remain stationary, whereas, for the MFT Challenge Disc, they were instructed to keep the balance board horizontal. If a participant lost their balance by stepping off the surface (e.g., from the MFT board), the test was restarted.

After each trial, participants were allowed to rest for one to three minutes as needed, but they were prohibited from standing on the unstable platform during these rest periods. At the end of each test, participants were asked to rate their perception of effort in completing the balance task using the Borg Scale for Rating Perceived Exertion (RPE) [42]. This scale measures exertion on a range from 0 to 10, where 0 indicates no perceived effort and 10 represents the maximum effort exerted [42].

### 2.3. Data Analysis

Acceleration data processing was performed using MATLAB^TM^ version R2024a (MathWorks Inc., Natick, MA, USA). Fourier analysis of the raw acceleration signals revealed the highest power in the frequency range of 5–10 Hz, with additional power observed in the 15–20 Hz range. Consequently, the signals were filtered with a 4th-order zero-phase 20 Hz low-pass Butterworth filter, as outlined in previous research [43,44]. Figure 2 presents examples of smoothed tri-axial acceleration data for bipedal balancing on a firm surface, a foam pad, and an MFT Challenge Disc plotted in three-dimensional coordinate space. For the acceleration-based variable analysis, the middle 100 s of the signals in the anteroposterior (AP), mediolateral (ML), and vertical (VT) directions were chosen for further analysis in order to exclude effects from initial adjustments and end-of-task restlessness [22]. 

Sample entropy (SampEn) was applied to the individual acceleration signals to assess the regularity of acceleration displacements, which reflects postural sway regularity and, consequently, postural control complexity [45,46]. The SampEn algorithm was calculated with three specific parameters: embedding dimension *m* = 2; tolerance *r* = 0.2·STD; where STD is the standard deviation of the time series; and a time delay (τ) = 100 ms, which corresponds to physiological timescales [32,47]. SampEn computes the probability that a sequence of data points, which has repeated itself within a tolerance r for a window length m, will also repeat itself for *m* + 1 points without allowing self-matches [45]. Higher SampEn values generally indicate greater postural control efficiency, reflecting a more adaptable postural control system [29,32], or suggesting less attentive, more automatic balance control [36].

### 2.4. Statistical Analysis

Statistical analyses were performed using SPSS software version 26.0 (IBM SPSS Statistics, SPSS Inc., Chicago, IL, USA), with a significance level set at α = 0.05. The normality of the variables was assessed using the Shapiro–Wilk test. To evaluate the impacts of age on participant characteristics, an independent sample *t*-test was employed.

A repeated-measures ANOVA, controlling for BMI, was conducted to examine how age and surface stability affect sample entropy (SampEn). The analysis assessed within-subjects effects of different surface stability conditions on SampEn values and between-subjects effects of age groups while adjusting for BMI to account for its influence. Effect sizes were reported using partial eta squared (η_p_^2^), and observed power (1-β) was calculated. Post hoc comparisons were performed with Bonferroni correction, adjusting the alpha level to 0.0167 for multiple comparisons. Additionally, Spearman correlations were used to explore relationships between SampEn and body composition variables.

The Friedman test was utilized to analyze the effects of surface stability on the perception of effort as measured by the Borg RPE scale, with pairwise comparisons between surfaces carried out using the Wilcoxon Signed Ranks Test, and their effect size (*r*) was reported. In addition, the Spearman correlation test between SampEn variables and perception of effort was also tested.

## 3. Results

### 3.1. Postural Control

All participants completed all balance tasks without losing stability or stepping to the floor for the foam board or balance board conditions. The age effects are observed in specific acceleration directions. Specifically, young adults show greater SampEn_ML value (F _(1,46)_ = 6.89, *p* = 0.012, η_p_^2^ = 0.130, 1-β = 0.729) and SampEn_AP value (F _(1,46)_ = 8.60, *p* = 0.005, η_p_^2^ = 0.157, 1-β = 0.819) than middle-aged adults (Figure 3A).

Additionally, the surface stability effects are observed in all acceleration directions: SampEn_ML (F _(2,92)_ = 19.29, *p* < 0.001, η_p_^2^ = 0.295, 1-β = 1), SampEn_AP (F _(2,92)_ = 11.40, *p* < 0.001, η_p_^2^ = 0.199, 1-β = 0.992), and SampEn_AP (F _(2,92)_ = 34.81, *p* < 0.001, η_p_^2^ = 0.431, 1-β = 1). However, pair-wise comparisons show significant differences in specific pairs of surfaces. As shown in Figure 3B, the increased instability level of the support surface decreases SampEn values in mediolateral (left) and anteroposterior acceleration (middle), while converse effects are observed in SampEn values in the vertical direction (right).

However, no interaction effects between age and surface stability factors are observed in SampEn variables of all acceleration directions. Spearman correlation analysis shows no correlation between all body compositions and all SampEn variables.

### 3.2. Perception of Effort

As shown in Figure 4, only the surface stability effects are observed for the perception of effort. Increasing instability levels of support surfaces progressively increase perceptions of effort (X^2^ (2) = 65.52, *p* < 0.001). Post hoc analyses show that bipedal balancing on a firm surface has lower effort than balancing on a foam surface (*p* = 0.001, effect size = −0.491) and an MFT Challenge Disc (*p* < 0.001, effect size = −0.840), respectively. Additionally, bipedal balancing on a foam surface has lower effort than balancing an MFT Challenge Disc (*p* < 0.001, effect size = −0.755). Spearman correlation analysis shows no correlation between the modified Borg Scale value and all SampEn variables.

## 4. Discussion

The current study aimed to investigate how age and surface stability affect postural stability by assessing the complexity of postural control through sample entropy (SampEn) of body acceleration in young and middle-aged adults during bipedal balance tasks on surfaces with varying levels of instability. The findings reveal significant age-related differences in postural control, as well as notable effects of surface stability on both postural control and perceived effort. However, no interaction effects between age and surface stability are observed. Both body composition variables and perception of effort did not influence the complexity of postural control.

Regarding age effects, the results show that younger adults had higher SampEn values in both the mediolateral (SampEn_ML) and anteroposterior (SampEn_AP) directions compared to middle-aged adults. These findings suggest that younger individuals have a more adaptable and efficient postural control system, as reflected by their greater postural sway complexity [29,32,35]. In contrast, middle-aged adults exhibited lower SampEn values, indicating a more rigid or less effective postural control strategy [29,32,35]. These findings challenge the previous notion that postural complexity in young and middle-aged adults during quiet standing may not differ significantly if the latter group is in good health [48]. The reduced postural control complexity observed in middle-aged adults might reflect the gradual effects of aging on neuromuscular control, such as sensory decline, which impairs the accuracy of perceiving and responding to postural disturbances due to slower and less precise muscle responses [49]. Moreover, since complexity is essential for adapting flexibly to changing environments, a lower complexity of postural control in middle-aged adults is indicative of reduced flexibility and increased rigidity in postural control strategies [34]. Although no statistically significant age effects were found in the perception of effort required to maintain balance, there was a trend toward higher perceived effort, suggesting that while neuromuscular control changes with age, this perception might not be as sensitive to age-related changes in postural control compared to other factors like surface instability. The increased perception of effort could be associated with the higher neuromuscular demands of balancing on unstable surfaces, which possibly requires more cognitive and muscular effort to compensate for decreased sensory accuracy and motor responsiveness with age [50]. Overall, the differences in postural control between younger and middle-aged adults appear to be mainly attributed to age-related changes in neuromuscular control rather than body composition (e.g., muscle mass) positively associated with muscle strength [51], indicating that the impact of aging on postural control is not significantly influenced by variations in body fat or muscle mass.

Additionally, the study found significant effects of surface instability on SampEn values across all acceleration directions. Three main points can be discussed. First, increased instability on support surfaces led to lower SampEn in the mediolateral direction, indicating that unstable surfaces challenge postural control and require more frequent adjustments or increased attention to balance [36]. This suggests that the person is using a more rigid and deliberate balance strategy to maintain stability. In other words, as surface instability increases, the body needs to focus more on making controlled and frequent adjustments to avoid losing balance, requiring more conscious effort and less automaticity in postural control, indicating more demanding, requiring increased neuromuscular coordination and attentional resources to maintain stability [36]. Second, higher sample entropy (SampEn) in the anteroposterior direction suggests greater irregularity and adaptability in postural control in that plane. This finding aligns with a typical postural control strategy, where the anteroposterior ankle sway is often the primary movement strategy used to maintain balance during bipedal standing on both stable and unstable surfaces [19]. In simpler terms, the body tends to rely on forward and backward ankle movements (anteroposterior sway) as the first line of defense to maintain stability when standing, indicating that the postural control system is more adaptable and capable of efficiently adjusting to balance disturbances, reflecting a more flexible and automatic balance control mechanism. This is particularly important on unstable surfaces, where adaptability is key to preventing falls [52]. Unstable surfaces reduce the usual sensory inputs needed to maintain balance, forcing the body to rely on less efficient adaptive control strategies [18], resulting in forcing the neuromuscular system to work harder, leading to increased muscle activation and more frequent postural adjustments to maintain stability [19,53]. Third, higher SampEn values in the vertical direction suggest greater irregularity or complexity in the control of postural sway, indicating that the neuromuscular system is more adaptable or less predictable when managing vertical perturbations. This increased complexity could imply that the system is engaging in more dynamic or less automated control strategies to maintain balance in the face of vertical disturbances, especially when the support surface becomes more unstable [36]. The vertical direction is less directly involved in typical balance strategies like anteroposterior (front-back) or mediolateral (side-to-side) sway, which are commonly used in maintaining postural stability [19]. Together, the current findings support the idea that unstable surfaces could facilitate neuromuscular demands, consistent with previous research [19,53]. Moreover, the perception of effort also increased with higher instability levels, with notable differences between firm, foam, and MFT Challenge Disc surfaces. Balancing on more unstable surfaces requires greater effort, reflecting the body expends more energy and the heightened demands on the neuromuscular system as observed in increasing myoelectric activities [19,53] and engages additional muscle groups to maintain postural stability [16].

The current findings have important implications for clinical practice. First, understanding how age and surface stability affect postural control can guide the development of targeted interventions for improving balance and preventing falls. Specifically, the observed decline in postural control complexity with age suggests that middle-aged and older adults may benefit from balance training that emphasizes adaptability and variability [54]. Incorporating balance training on unstable surfaces could enhance postural control by challenging and improving the neuromuscular system’s ability to respond to dynamic changes, thus potentially reducing fall risk [55,56]. Second, surface instability plays a crucial role in enhancing proprioception by requiring the body to make continuous adjustments to maintain balance. This instability stimulates proprioceptive feedback mechanisms, which are essential for detecting changes in joint position and muscle tension [57,58]. Muscle spindles are key proprioceptors that increase their sensitivity to rapid changes in muscle stretch on unstable surfaces, providing detailed feedback about the body’s position and movement. This enhanced feedback helps the central nervous system adjust muscle contractions more precisely, improving postural control. Additionally, mechanoreceptors such as Ruffini endings, Pacinian corpuscles, and Golgi tendon organs contribute by modulating their response to detect subtle shifts in surface stability and muscle tension [58]. Finally, recognizing the impact of surface stability on perceived effort underscores the need for exercise programs that balance effectiveness with comfort.

In summary, younger adults demonstrate greater postural control complexity and adaptability compared to middle-aged adults, who exhibit more rigid control strategies. Surface instability significantly impacts postural control and perceived effort, with increased instability leading to greater demands on the neuromuscular system. These findings highlight the need for tailored rehabilitation strategies that address age-related changes and surface stability challenges to improve balance and prevent falls.

### Limitations and Future Research

This study has several limitations that should be considered. First, although the sample size of 48 participants exceeds the minimum required by power analysis (*n* = 36), it remains relatively small. This may affect the detection of subtle effects and the generalizability of the findings. Larger sample sizes in future studies would enhance the robustness and reliability of the results. Second, participants self-reported having no neurological or musculoskeletal issues and no balance-specific training in the past six months. While this helps control some variables, it may not account for all individual differences, potentially limiting the applicability of the findings to individuals with different health conditions or balanced training histories. Third, the cross-sectional design limits the ability to draw conclusions about causality or changes over time. A longitudinal approach could provide more insights into how age and surface instability affect postural control over extended periods. Fourth, while SampEn is valuable for assessing postural control complexity, it may not capture all aspects of balance. Future studies should incorporate additional measures, such as electromyography (EMG), to assess muscle activation patterns to gain a more comprehensive understanding of balance mechanisms [59]. Lastly, although body composition was assessed, no significant differences were found between age groups. This suggests that the differences in postural control are more likely due to age-related changes in sensory, motor, and cognitive functions rather than variations in body composition.

## 5. Conclusions

The current study examined the effects of age and surface stability on the complexity of postural control assessed by sample entropy, focusing on the transition from young adulthood to middle adulthood. The results indicate that younger adults exhibit greater postural control complexity, as evidenced by higher SampEn values in both mediolateral and anteroposterior directions, compared to middle-aged adults. No significant age-related differences in body composition were observed, suggesting that body composition does not play a primary role in the observed age effects on postural stability. The more adaptable and efficient postural control in younger adults contrasts with the more rigid control strategy seen in middle-aged adults. Additionally, increased surface instability was found to decrease SampEn values in the mediolateral direction, reflecting the greater demands placed on the neuromuscular system to maintain balance on unstable surfaces. These findings highlight the importance of tailored rehabilitation and exercise programs to support balance, especially for middle-aged adults.

## Figures and Tables

**Figure 1 sensors-24-06846-f001:**
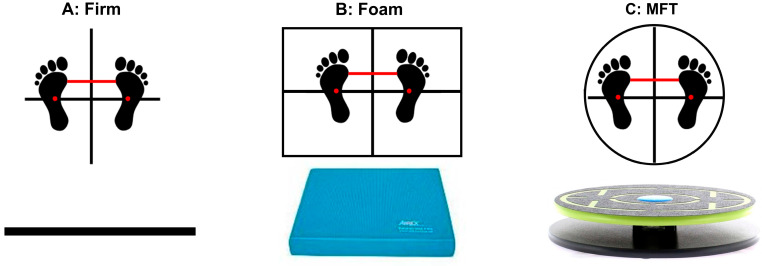
Starting position of bipedal balancing on: (**A**) a firm surface, (**B**) a foam pad, and (**C**) an MFT Challenge Disc. Note: the horizontal red line between the feet (the medial borders of each distal end of the first metatarsal bone) is aligned with an individual inter-feet distance (15% of biacromial diameter).

**Figure 2 sensors-24-06846-f002:**
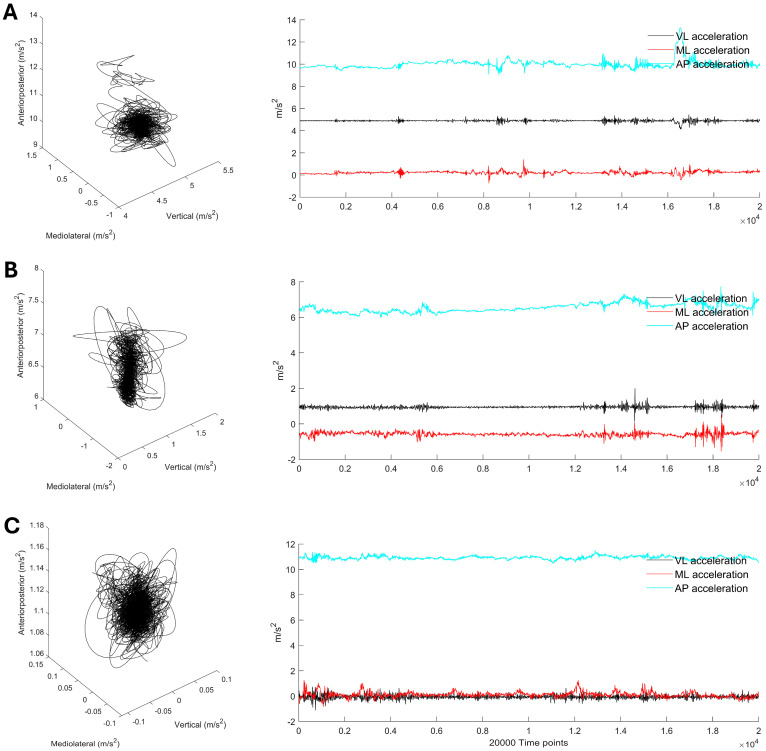
Examples of tri-axial acceleration displacements along with their corresponding time series data retrieved from bipedal balancing on (**A**) a firm surface, (**B**) a foam pad, and (**C**) an MFT Challenge Disc, with the corresponding sample entropy (SampEn) calculated for each sway direction. Note: the data presented were obtained from one of the middle-aged participants during the middle 100 s (20,000 time points) of bipedal balancing on each support surface.

**Figure 3 sensors-24-06846-f003:**
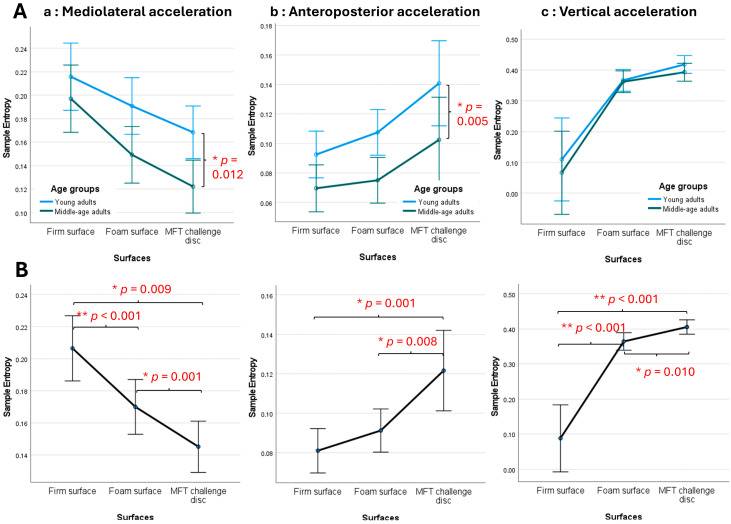
Effects of (**A**) age and (**B**) surface stability on sample entropy of (**a**) mediolateral, (**b**) anteroposterior, and (**c**) vertical accelerations (group mean ± standard error, * *p* ≤ 0.012, and ** *p* < 0.001).

**Figure 4 sensors-24-06846-f004:**
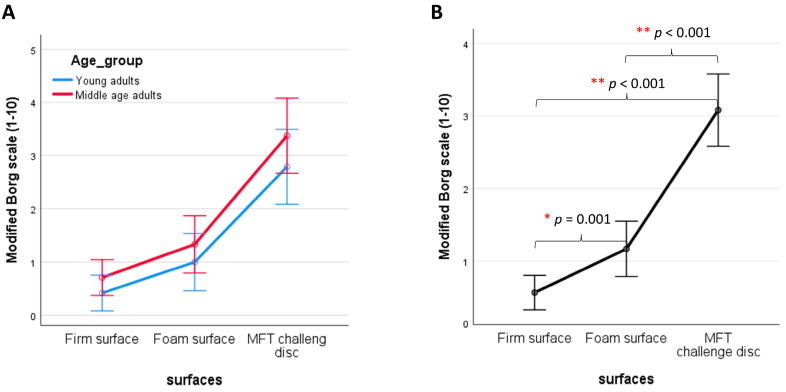
Effects of (**A**) age and (**B**) surface stability on the perception of effort, as measured by the 10-point Borg Scale for Rating of Perceived Exertion (RPE). Data are presented as group means ± standard error. Statistical significance is indicated as * *p* = 0.001 and ** *p* < 0.001.

**Table 1 sensors-24-06846-t001:** Characteristics of participants (mean ± SD); * *p* < 0.05).

	Young (*n* = 24)	Middle Age (*n* = 24)	*p*-Value	Effect Size
Age (years)	23.9 ± 5.3	51.4 ± 5.9	<0.001 *	4.903
Weight (kg)	51.4 ± 10.0	59.8 ± 10.5	0.088	0.819
Height (cm)	156.9 ± 4.5	153.5 ± 8.7	0.094	0.490
Body mass index (BMI, kg/m^2^)	22.2 ± 4.2	25.6 ± 5.1	0.016 *	0.727
Body fat (%)	31.6 ± 5.9	34.6 ± 5.8	0.088	0.512
Fat-free mass (%)	68.4 ± 5.9	65.4 ± 5.8	0.088	0.512
Muscle mass (%)	36.2 ± 3.7	36.2 ± 5.5	0.988	0.000

## Data Availability

The data supporting this study is available from the corresponding author upon reasonable request.

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
