# Peer review of "Age-Related Changes in Postural Stability in Response to Varying Surface Instability in Young and Middle-Aged Adults"

_sensors, 2024, doi:10.3390/s24216846_

Round 1
Reviewer 1 Report
Comments and Suggestions for Authors
It is an interesting topic, especially with regard to rehabilitation and fall prevention in older adults. While it is true that longitudinal studies are generally more relevant for this type of research, I believe the results provide valuable information for advancing the topic.
I suggest that the authors analyze the differences in each of the stability conditions independently, rather than as a whole, based on age. Additionally, if sex influences postural control, why was it not included as a variable in the study? The authors should consider incorporating these methodological suggestions.
The introduction is too long. It would be advisable to summarize it and focus only on the key aspects related to the study’s topic.
Regarding Figure 4B, the explanation is unclear. Does it refer to the perceived effort of both age groups? If so, why wasn't this presented independently, as was done in Figure 4A?
Line 346: Does the summary in these lines form part of the conclusions? It seems somewhat redundant.
Author Response
Comment: I suggest that the authors analyze the differences in each of the stability conditions independently, rather than as a whole, based on age. Additionally, if sex influences postural control, why was it not included as a variable in the study? The authors should consider incorporating these methodological suggestions.
Response: Thank you for your suggestion. We recognize the importance of exploring the influence of sex and surface stability as potential factors affecting postural control. However, analyzing the differences in each stability condition independently is beyond the current study's objectives. We apologize for any oversight regarding this aspect.
Comment: The introduction is too long. It would be advisable to summarize it and focus only on the key aspects related to the study’s topic.
Response: Thank you for your feedback regarding the introduction. We believe that retaining all supportive information is essential for enhancing the clarity of the study's objectives for the audience.
Comment: Regarding Figure 4B, the explanation is unclear. Does it refer to the perceived effort of both age groups? If so, why wasn't this presented independently, as was done in Figure 4A?
Response: We appreciate your observation about Figure 4B. This figure presents the average perceived effort for both age groups based on the statistical analysis (ANOVA) used in the current study. We will clarify this in the manuscript.
Comment: Line 346: Does the summary in these lines form part of the conclusions? It seems somewhat redundant.
Response: Thank you for pointing out the potential redundancy in the summary on line 346. We have reviewed this section and removed the redundancy.
Reviewer 2 Report
Comments and Suggestions for Authors
The manuscript “Age-Related Changes in Postural Stability in Response to Varying Surface Instability in Young and Middle-Aged Adults” describes the study of balancing ability to maintain posture in complex conditions for 48 participants, including 24 young and 24 middle-aged adults.
As the authors note, postural impairment is linked to age-related sensory deficits, and such studies are essential for creating effective strategies to improve postural stability and prevent falls. The results are consistent and support this widely accepted concept.
Plots in the Figure 2 show the 3D trajectories in the acceleration space with each axis corresponding to the acceleration in m/s^2.
1. Average acceleration seems to be non-zero. For instance, anteroposterior acceleration is always greater than 1 m/s^2 for Figure 2 C, which would give the displacement after 2 minutes of more than S = a * t^2/2 = 1 * 120 * 120 / 2 m = 7200 m. This issue should be resolved.
2. Related to the above: Individual plots of acceleration along each axis versus time would be interesting to see, as well as the displacements.
Author Response
The manuscript “Age-Related Changes in Postural Stability in Response to Varying Surface Instability in Young and Middle-Aged Adults” describes a study examining the balancing ability to maintain posture under complex conditions for 48 participants, including 24 young and 24 middle-aged adults. As the authors note, postural impairment is linked to age-related sensory deficits, making such studies essential for developing effective strategies to improve postural stability and prevent falls. The results are consistent and support this widely accepted concept.
Response: Thank you for your thoughtful and constructive feedback on our manuscript. We greatly appreciate your comments and suggestions for improvement.
The plots in Figure 2 show the 3D trajectories in the acceleration space, with each axis corresponding to acceleration in m/s².
- Comment: You noted that the average acceleration appears to be non-zero. For instance, in Figure 2C, the anteroposterior acceleration is consistently greater than 1 m/s², which would imply a displacement of more than than S = a * t²/2 = 1 * 120 * 120 / 2 m = 7200 m after 2 minutes. This issue needs to be resolved.
Response: Thank you for your careful observation. Upon reviewing the data and Figure 2, I realized that I had incorrectly reported the unit of acceleration in the figure labels. The measurements should have been expressed in mm/s² rather than m/s². In our study, we used a scaled-down version of the SI unit for acceleration (1 m/s² = 1000 mm/s²) to reflect the small accelerations involved in balance tasks.
Since all participants were healthy and instructed to stand still on firm and foam surfaces and maintain the board flat during the MFT challenge disc, it would be impossible for the balancing movements to exceed 1 m/s². I have corrected the unit in the manuscript and updated Figure 2 to accurately represent the 3D trajectories of acceleration.
- Comment: Related to your comment, you suggested that individual plots of acceleration along each axis versus time, as well as the displacements, would be interesting to see.
Response: I appreciate your suggestion to include these additional plots. I have incorporated individual plots of acceleration along each axis versus time, as well as the displacements, into the manuscript to provide a more detailed and clear representation of the data.
Thank you once again for highlighting these issues, which have significantly improved the clarity and accuracy of our manuscript.
Reviewer 3 Report
Comments and Suggestions for Authors
This study aims to investigate how age and surface stability affect postural in young and middle-aged adults during bipedal balance tasks on surfaces with varying levels of instability. The study is well written, the methodology sounds, the results are clearly presented.
I have only one minor comment, as I think this reference could be valuable and should be added in the introduction with reference 6 : dot 10.1159/000520959.
Author Response
This study aims to investigate how age and surface stability affect postural control in young and middle-aged adults during bipedal balance tasks on surfaces with varying levels of instability. The study is well written, the methodology is sound, and the results are clearly presented.
Comment: I have only one minor comment, as I think this reference could be valuable and should be added in the introduction with reference 6: dot 10.1159/000520959.
Response: Thank you for your positive feedback and for highlighting the reference suggestion. I appreciate the opportunity to improve the manuscript further. I have reviewed the recommended reference and agree that it provides valuable insight relevant to the study’s focus on age-related changes and the risk of falls. I have incorporated this reference into the introduction alongside reference 6 to strengthen the background discussion on the impact of aging on postural stability.
Round 2
Reviewer 2 Report
Comments and Suggestions for Authors
The paper has been improved. Still, I cannot understand why the mean acceleration is non-zero.
Probably there is a zero offset of the accelerometer.